# Chondroitin Sulfate Safety and Quality

**DOI:** 10.3390/molecules24081447

**Published:** 2019-04-12

**Authors:** Nicola Volpi

**Affiliations:** Department of Life Sciences, Laboratory of Biochemistry and Glycobiology, University of Modena and Reggio Emilia, 41125 Modena, Italy; nicola.volpi@unimore.it; Tel.: +39-59-2055543; Fax: +39-59-2055548

**Keywords:** chondroitin sulfate, glycosaminoglycans, osteoarthritis, nutraceuticals, food supplements

## Abstract

The industrial production of chondroitin sulfate (CS) uses animal tissue sources as raw material derived from different terrestrial or marine species of animals. CS possesses a heterogeneous structure and physical-chemical profile in different species and tissues, responsible for the various and more specialized functions of these macromolecules. Moreover, mixes of different animal tissues and sources are possible, producing a CS final product having varied characteristics and not well identified profile, influencing oral absorption and activity. Finally, different extraction and purification processes may introduce further modifications of the CS structural characteristics and properties and may lead to extracts having a variable grade of purity, limited biological effects, presence of contaminants causing problems of safety and reproducibility along with not surely identified origin. These aspects pose a serious problem for the final consumers of the pharmaceutical or nutraceutical products mainly related to the traceability of CS and to the declaration of the real origin of the active ingredient and its content. In this review, specific, sensitive and validated analytical quality controls such as electrophoresis, eHPLC (enzymatic HPLC) and HPSEC (high-performance size-exclusion chromatography) able to assure CS quality and origin are illustrated and discussed.

## 1. Chondroitin Sulfate Structure

Hyaluronic acid (HA), keratan sulfate (KS), chondroitin sulfate (CS)/dermatan sulfate (DS) and heparan sulfate (HS)/heparin are natural macromolecules and the main complex heteropolysaccharides belonging to the class of glycosaminoglycans (GAGs) [1,2]. Apart from KS, which is composed of galactose and *N*-acetyl-d-galactosamine (GalNAc) [3], the backbone of the other GAGs is commonly represented by typical disaccharide sequences constituted of alternating units of uronic acids, glucuronic (GlcA) or iduronic (IdoA), and amino sugars, *N*-acetyl-d-glucosamine (GlcNAc) or GalNAc [4]. KS, CS/DS and HS/heparin are sulfated macromolecules possessing different degrees of charge density thanks to the presence of sulfate groups in varying amounts and located in different positions. Moreover, besides the variability in sulfate groups, these polysaccharides are very heterogeneous in terms of molecular mass, physicochemical properties, and biological and pharmacological properties [1,2,3,4].

CS with different disaccharides and overall carbohydrate backbones may be biosynthesized possessing various number and position of sulfate groups [4] (Figure 1). Consequently, CS is a largely heterogeneous GAG formed of alternate and variously sulfated disaccharides of GlcA and GalNAc linked by β(1→3) bonds. Moreover, the different disaccharides are linked to each other by β(1→4) bonds. CSA and CSC are constituted by disaccharides sulfated in position C4 or C6 of the GalNAc residue, respectively, and minor percentages of the monosulfated disaccharide on C2 of GlcA are possible. Besides the main presence of these two kinds of disaccharide units monosulfated in position C4 or C6 of GalNAc, disaccharides with a different number and position of sulfate groups can be present, in various percentages, within the carbohydrate chains [4,5,6]. Various kinds of disulfated disaccharides may be present in the CS structure in various amounts also in relation to specific animal sources (as in cartilaginous or bony fishes, see below), such as the disaccharide disulfated in position C2 of GlcA and C6 of GalNAc (the so-called disaccharide D), in position C4 and C6 of the GalNAc unit (the disaccharide E) or C2 of GlcA and C4 of GalNAc (the disaccharide B) (Figure 1). Finally, a low percentage of the non-sulfated disaccharide is almost always present in the CS backbone while the fully tri-sulfated disaccharide is generally in trace amount [4,5,6].

Besides the above-illustrated CS structures, a peculiar CS polysaccharide known as CSB or DS is known in nature, formed of disaccharide units of IdoA and GalNAc mainly sulfated in position C4 with a minor and variable content of the disulfated disaccharide in position C4 of GalNAc and C2 of IdoUA [4]. However, the biosynthetic processes generally produce highly modified oligosaccharide domains separated by regions having a low-degree of structural modifications introducing further heterogeneity. Consequently, the CS and DS chains may be hybrid polymers of low modified (CS) and highly modified (DS) domains [4]. The sulfation heterogeneity introduces a great variability in the position of sulfate groups and in CS charge density. Moreover, the different number of disaccharide units forming the CS polymer and the overall molecular weight generated are other key factors influencing its properties. Finally, the combination of the different sulfated disaccharides and oligosaccharide domains produces a huge number of heterogeneous CS polymers possessing various, specific and specialized biological and pharmacological functions [1,2].

## 2. Chondroitin Sulfate Activity

CS is a natural biomacromolecule abundantly distributed in virtually all invertebrates and vertebrates (and consequently in humans) and involved in many biological processes [1,5,7,8]. Based on its organism-to-organism and tissue-to-tissue structural diversity in chain length and sulfation patterns, CS provides specific biological functions at molecular, cellular and organ level such as cell adhesion, cell division and differentiation, morphogenesis, organogenesis and neural network formation [9,10,11]. Moreover, CS is particularly abundant in all mammalian connective tissues, especially in the cartilage, skin, blood vessels, ligaments and tendons with different structure and level of sulfation [12]. In addition to its conventional structural roles in the composition of extracellular matrix and formation of organs and tissues, such as cartilages and bones, recent evidence demonstrates that CS chains are involved in important biological functions in inflammation, infection and wound repair [7,8,9,10,11].

The biological effects of CS are related to its ability to interact with a wide variety of other bio(macro)molecules including (but not limited to) matrix molecules, growth factors, protease inhibitors, cytokines, chemokines, adhesion molecules and pathogen virulence factors via aspecific/specific sulfated saccharide domains within the chains [11,12,13]. In fact, along with aspecific interactions due to its numerous negative charges provided by the sulfate and carboxyl groups, CS is involved in specific binding to bioactive molecules for the presence of peculiar functional domain structures which are formed by combinations of the various disaccharide units [11,12,13] (see Figure 1).

Osteoarthritis (OA) is a global issue affecting all known countries and ethnicities with huge incidence and economic impact [14]. OA particularly affects the older population producing variable degrees of limitation of movement and major activities of daily living. Moreover, OA is expected to dramatically rise in the future, in particular for the population above 70 years of age [14,15,16], also related to the observed increase of obesity in western populations [17,18]. Current treatments are mainly based on pain reduction [19] and treatments having the aim to reduce the progression of the disease, such as the S/DMOADs (structure/disease modifying anti-osteoarthritis drugs) [20,21]. CS inhibits the extracellular proteases involved in the metabolism of connective tissues, and it is able to stimulate proteoglycans production by chondrocytes as well as reduce the cartilage cytokine production but also to induce apoptosis of articular chondrocytes [8,22]. Furthermore, CS increases the intrinsic viscosity of the synovial liquid and, in bones, it accelerates the mineralization process and repair [8,22]. Overall, CS has a key role in articular and bone metabolism by controlling cartilaginous matrix integrity and bone mineralization. Consequently, with others, treatment with CS is reported to improve OA symptoms and to reduce the disease. In fact, in many published studies, CS has demonstrated to be a symptomatic slow acting drug for the treatment of OA (SYSADOA) and to have disease modifying properties in long-term treatments (so-called DMOAD) (see [20,21,22,23,24] for reviews).

## 3. Safety and Quality Concerns of Chondroitin Sulfate

CS, according to its nature, must be extracted from animal sources and submitted to purification processes for commercial purposes [5,6]. Actual commercial production of CS is based on terrestrial raw material, from bovine, porcine, and chicken, or marine sources such as cartilaginous fish, sharks and skate, but also bony fishes ([5,6,25] and personal information). Moreover, a mix of all these sources is possible, producing a CS final product with mixed characteristics and properties. The use of possible not controlled raw animal material (tissues, bones and cartilages but also soft organs) poses the problem of a final product with not controlled structure and poor reproducibility along with the not well identified origin with the consequence of variable grade of purity, biological effects, presence of contaminants, clinical efficacy and safety [5,6,26].

Besides the source material, manufacturing processes, the presence of contaminants and other natural biomolecules, as well as many other factors, contribute to the quality, structure and properties of the final product influencing CS overall biological, nutraceutical and pharmacological capacities. In fact, as well known, CS possesses a complex structure strictly depending on the tissue, organ and species but also on the age of the animals [5,6]. Moreover, the biological and nutraceutical properties may vary with the structure as well as the oral absorption that is influenced by the different CS physicochemical properties [27,28,29]. Furthermore, due to the extractive origin of CS and to the different purification processes applied, the presence of bacteria, virus or prions cannot be excluded [30]. Additionally, other various natural bioactive (macro)molecules may be present in different content as contaminants in CS final extracts [5,6], and voluntary adulteration by artificial compounds is also known. Finally, a restriction use related to religious issues is possible (see below for these CS concerns). Overall, depending on the source origin and on the manufacturing processes, variable CS final products for structure, quality and activity may be produced (Table 1) [5,6,26].

CS from various sources has a carbohydrate backbone formed of disaccharides with sulfate groups in different percentages and positions (see Figure 1) producing polymers possessing different charge densities. Furthermore, due to the biosynthetic processes, CS macromolecules with different grades of polymerization may be generated with various molecular masses and polydispersity related to specific tissues, organs and species. Because of these structural variations, and the further possible presence of specific oligosaccharide sequences (and purity, see below) of the preparations for therapy applications or in nutraceuticals, CS may have different biological properties and capacities [5,26]. According to this structural variability, different and peculiar activities have been reported depending on the CS structure [5]. Finally, bioavailability and pharmacokinetic have been reported to change depending on CS structural properties and origin when orally administered during therapy [27,28,29].

Extraction and purification processes may further produce CS preparations having different structures introducing modifications of the macromolecular characteristics and chemical properties by controlled as well as undesired depolymerization, desulfation and/or chemical modifications [6].

Another key point is related to the accurate evaluation of the CS content and purity when the preparation is produced for therapeutic or nutraceutical applications. In fact, the CS content and purity may change according to the manufacturing processes and/or tissue sources. Depending on the purification protocols, the extracts may have a variable content and purity of CS due to the presence of polluting inorganic or organic side-products (Table 1).

All the above-mentioned aspects pose a serious problem for the pharmaceutical or nutraceutical Companies having the necessity to declare the real origin of the active ingredients and the label traceability of CS from the raw material up to the finished products. Moreover, an accurate label indication of the CS content is of paramount importance to assure to the final consumer a real dose administration also considering the possible intentional adulteration by potentially dangerous (macro)molecules (see below) mimicking CS structure and properties.

### 3.1. Natural Contaminations

As already mentioned, commercial CS produced from animal sources requires long and complex procedures of extraction and purification with the aim to eliminate or reduce the content of the other (macro)molecules present as natural contaminants [26] (Figure 2). At the same time, contamination of CS may be accidentally caused by the same extraction and purification processes if not well controlled. Obviously, as more steps of purification are adopted having differential capacities in the elimination of natural contaminants of various nature, CS final products with higher quality may be obtained. On the other hand, the production of high-quality CS of pharmaceutical grade generally requires high costs of production and quality analytical controls.

#### 3.1.1. Polysaccharides

Depending on the animal source and on the extraction and purification procedures that generally are complex and require long time, CS extracts and final products for commercial purposes are also polluted with variable percentages of other polysaccharides that are co-extracted with CS during its preparation (Figure 3).

HA in particular, and DS with its carbohydrate backbone made of iduronic acid, have been detected in raw materials and formulations and reported in scientific papers [31] due to their ubiquitous presence in tissues from which CS is generally produced. Moreover, KS has been detected in many batches of CS produced from shark cartilage, averaging 16% of the total GAGs [32,33] (Figure 4). Indeed, as KS is involved in the structure of the proteoglycan aggrecan located in all cartilages, it has been detected in extracts and finished preparations of various origin along with CS extracted from the same cartilages [34]. A total of 15 samples, including four samples of CS as laboratory reagents, one sample of CS as a food additive and ten samples of dietary supplements containing CS, were examined for the presence of KS by using specific immunodiffusion and enzyme-linked immunosorbent assay (ELISA). Apart from the three samples of CS as laboratory reagents, all samples were found to contain varying amounts of KS [34]. Finally, it is worth mentioning that KS possesses immunogenic capacities able to develop immune reactions [35]. Depending on the animal raw material used for CS extraction, HS may also be present due to its comparable physicochemical properties with CS, in particular for its nature of sulfated heteropolysaccharide [12].

The presence of these natural polymers in CS products, also in high percentages, alerts the manufacturers for improved isolation procedures as well as the supervisory agencies for better audits. In addition, this finding also compromises the desired amounts of the real ingredient specified on the label claims and forewarns the pharmacopeias to update their monographs. In fact, general non-specific analytical methods such as CPC (cetylpyridinium chloride) titration and carbazole assay are unable to detect the presence of other polysaccharides in CS products due to their quite similar properties and chemical structures (see below for further analytical considerations). Finally, the presence of variable amounts of the immunogenic KS poses serious problems related to the possible development of immunogenic adverse reactions (Figure 3).

#### 3.1.2. Other Biomolecules

Other (macro)molecules possessing similar properties of CS are copurified during CS extraction from tissues, such as nucleic acids [31] and proteins (Figure 5). In fact, animal tissues and organs are rich in DNA, RNA and proteins, and variable content of these biomolecules are present in CS extracts depending on the source and on the purification protocols adopted. Moreover, some of these proteins may have allergenic and/or intolerance capacity able to develop immune reactions. Finally, these macromolecules also give positive response to aspecific analytical controls such as CPC assay [6] (Figure 5).

#### 3.1.3. Infective Agents

The animal origin of CS may pose safety problems due to the possible presence of pathogen contamination and transmissible infective agents such as bacteria residues, viruses and prions potentially causing spongiform encephalopathy in bovines (BSE), foot-and-mouth disease, influenza spread in birds and other animal diseases (Figure 5). The possible presence of infective agents in the organs and tissues used for CS production requires the application of specific chemical steps able to destroy or inactivate and eliminate bacteria, viruses and prions [36,37] possibly introducing chemical modifications in the CS structure such as degradation, desulfation, oxidation of carbohydrate chains, introduction of chemical groups, etc. In fact, prions adhere very tenaciously to surfaces, making them hard to remove, and their resistance to protease treatment, certain chemical agents and heat denaturation represents a major concern for pharmaceutical products derived from terrestrial animals [36]. Moreover, the purification steps useful to remove the infective agents can also affect the biological capacities of extracted CS (Figure 6).

Another key point is the adoption of specific and selective analytical assays to detect the possible presence of the different infective agents in the final products. In fact, the presence of prions may be easily demonstrated with a diagnostic test. However, the current available analytical assays are characterized by high costs and limited to specialized laboratories [38]. Finally, other pathogens show similar attributes as prions for the potential risk of carry-over into final purified products.

### 3.2. Intentional Adulterations

Besides the presence of natural contamination, finished CS preparations may also be adulterated by the voluntary addition of cheaper polysaccharides and organic derivatives unable to be determined with usual analytical controls utilized in the nutraceutical industry (Figure 7). In fact, CS is well known to be one of the most adulterated supplements in the market, a widespread and probably underestimated problem, abetted by the lack of industry-wide specific and sensitive quality control analytical methods with several of the adulterated agents going undetected [6,39].

Approximately 80–90% of the CS supplied to the US and Europe markets is produced in China and intentional adulteration is possibly practiced by nutraceutical manufacturers globally with the aim to reduce ingredient costs by using cheaper substances [40]. The price pressure is also responsible for poor purification procedures with the production of CS preparations rich in salts and having low title but also to minimize analytical controls to only common non-specific and inexpensive tests [40]. In this scenario, some raw material and dietary supplement products contain less than the claimed amount of CS, in some cases as little as 5–10% [41,42] with a huge fraud to clients, traders and consumers (Figure 7). In fact, many studies available in scientific literature demonstrate the poor quality and a title not conforming to label declaration of many CS finished products in several countries [6,41,42,43,44,45].

The application of specific analytical methods such as eHPLC (enzymatic HPLC), HPSEC (high-performance size-exclusion chromatography) and electrophoresis (Figure 4 and see below for analytical methods) on commercially available nutraceuticals has demonstrated the poor quality of many of these products in the global market showing a lower concentration of CS than specified on the label as well as their intentionally adulteration [39,41,42]. The adulteration is obtained with molecules that may interfere with aspecific widely-used analytical methods of CS quantitation, such as the CPC titration and carbazole assay, to give an artificially inflated estimate of CS concentration [6]. Many (macro)molecules have been identified for CS adulteration, such as carrageenan, proteins and surfactants, cheaper polysaccharides, sodium alginate, propylene glycol alginate sulfate sodium, sodium hexametaphosphate commonly known with the name of Calgon and used as a detergent or water-treatment additive [39], maltodextrin and lactose [42] (Figure 7). These substances can be separated from real CS by their difference in electrophoretic mobility (Figure 4) and a correct title of CS may be obtained by other specific analytical approaches such as eHPLC and HPSEC (see below), and not with commonly-used methods [6,39,41,42] (Figure 7).

## 4. Analytical Quality Controls

CS for pharmaceutical applications is strictly evaluated for quality, content, structural characterization and parameters (such as charge density and molecular mass) by means of sensitive, specific, validated and published analytical approaches [6]. These features are of paramount importance to ensure therapeutic reproducibility and safe use, and to protect patients from ineffective and/or unsafe and potentially dangerous drugs. As discussed below, is it possible to affirm the same for food and nutraceutical CS preparations?

European and USA pharmacopoeias require some analytical tests for CS identification such as infrared spectroscopy, specific optical rotation, intrinsic viscosity, electrophoresis and CPC titration [46,47]. Some of these assays are very old and usually no longer used in modern quality control laboratories due to their lack of specificity, low sensitivity and poor reproducibility. In particular, infrared spectroscopy and specific optical rotation cannot give specific information belonging to many other (macro)molecules, such as the absorption of chemical groups in the spectrum of infrared wavelengths (for infrared spectroscopy) [48] or the capacity to rotate the plane of polarization of plane-polarized light by chiral chemical compounds such as carbohydrates (for optical rotation) [49]. Intrinsic viscosity measurement is a very old technique for the determination of the molecular weight of a polymer replaced by the more modern size-exclusion chromatography coupled with various detectors [50] (Figure 8). The industry-wide method for analysis of CS is CPC titration. However, several flaws in this old testing method show relevant limits, as it can be tricked by the use of natural macromolecules and/or artificial adulterants and cannot distinguish between CS and other material having similar physicochemical properties, e.g., nucleic acids, anionic proteins, other polysaccharides, etc. [6]. Moreover, CPC assay is also influenced by the molecular weight of tested CS as well as for the carbazole assay that is another non-specific quantitative test [6]. On the contrary, agarose-gel electrophoresis according to European pharmacopoeia, electrophoresis on acetate of cellulose (ACE) according to US pharmacopoeia, and HPSEC and eHPLC according to the AOAC (Association of Official Analytical Chemists) [51] are highly specific analytical approaches for the determination of CS quality, quantity, chemical properties and structure.

Besides the low title, also origin of CS has the been reported to be a puzzle and in general not declared on product labels [6]. The origin of the CS utilized in a finished product can only be detected by rather expensive and detailed biochemical/structural analyses by determining the disaccharide/oligosaccharide pattern and molecular mass parameters [6]. According to this, a combination of specific analytical methods to define the final quality, origin and properties of the product is required. Such methods include electrophoresis, HPSEC, selective enzymatic treatment and further separation of the constituent disaccharides/oligosaccharides using sophisticated techniques including capillary electrophoresis [31,52,53], HPLC [5,6,54,55], FACE (Fluorophore-Assisted Carbohydrate Electrophoresis) [42,56] and mass spectrometry [57]. CS profile is achieved using well-established multi-analytical laboratory techniques, particularly electrophoresis, HPSEC and eHPLC (Figure 8) [5,6] due to their high throughput, sensitivity and reproducibility. These techniques, applied before and after enzymatic digestion of CS into smaller components, can distinguish between CS derived from different animal or marine sources based on disaccharide content, patterns of sulfation and molecular size of the polymer (Figure 8) also using analytical standards of CS of different origin (see below). However, a big problem remains for the sure identification of the origin of CS when mixed source material is used to its production or in the case of possible cross-contamination.

The use of animal-derived sources to produce commercial CS also represents a concern for vegetarians and people with dietary restrictions related to religious and supply issues. In fact, religious reasons and/or the practice of abstaining from the use of animal products have precluded the introduction of CS dietary supplements both in key emerging markets, such as Middle East and Asia, and in consolidated markets.

The application of the above-illustrated multi-analytical techniques, electrophoresis, HPSEC and eHPLC requires highly qualified laboratories equipped with dedicated instrumentation and qualified technicians with specific analytical know-how producing very expensive activity and elevated costs for high quality CS management (Figure 8). Moreover, an accurate and reproducible quantitative assay requires reference standards with high specificity, purity and well-known physicochemical properties and structures [6]. This is not a simple task in the case of CS due to its high heterogeneity of structure and properties mainly depending on the source and purification protocols, as previously discussed. However, several reference standards are commercially available, such as the CS standard from bovine cartilage manufactured by Bioiberica (www.bioiberica.com/) and possessing a purity greater than 98% and approved in 2004 as Chemical Reference Substance (CRS) by the European Pharmacopeia Commission [6]. Furthermore, other standards are available such as the European pharmacopoeia reference standard of CS sodium of marine origin (https://www.sigmaaldrich.com/) or the CS sodium salt from shark cartilage (https://www.sigmaaldrich.com/).

## 5. Conclusions

In 2008, heparin manufactured in China was intentionally adulterated with an oversulfated CS polymer having a very high charge density quite similar to natural heparin, which provoked hundreds of deaths in Europe and USA after intravenous administration [58]. As for heparin, an important natural drug derived from animals, intentional adulteration of CS is difficult to detect since the tissue supply chain for CS in slaughterhouses generally lacks current good manufacturing processes (cGMP) oversight assuring high-quality CS production along with specific analytical quality controls. Moreover, as mentioned above, CS is derived from a variety of animal tissues potentially containing infectious agents leading to the transmission of viral, bacterial and prion diseases and the intentional or accidental mixing of CS from different animal species is highly suspected. In fact, some countries do not have a strict biosecurity system able to reduce the risk of the introduction and spread of disease agents.

Even if specific and high-level analytical methods are generally able to distinguish from various sources, the detection of the presence of small amounts of different raw material remains very difficult. Additionally, CS structure can vary with environmental factors, animal subspecies, age and/or tissues providing additional difficulties for its quality and properties standardization. Moreover, the missing guidelines allow manufacturers of finished products to often change the sources of supply of the animal tissues or CS raw materials for competitive advantages. Consequently, the urgent solution to this scenario is the introduction and utilization in the current market of validated, specific, scientifically sound and largely published analytical controls. Manufacturers should ensure traceability of raw material right through to the finished product, with analytical evaluation made of each lot supplied to guarantee same quality, reproducibility and safety, as well as avoid misleading label information on finished products as reported in the literature.

In conclusion, specific and accurate analytical procedures should be enforced for the control of high-quality products and applied by quality control laboratories to confirm the purity and label claims of CS in raw materials and finished products for food and nutraceutical applications. Finally, there are no definite regulations and common acceptance governing the origin of the ingredients in these supplements and the origin of the ingredients in products of natural and extractive origin is one of the most important factors ensuring quality, safety and efficacy.

## Figures and Tables

**Figure 1 molecules-24-01447-f001:**
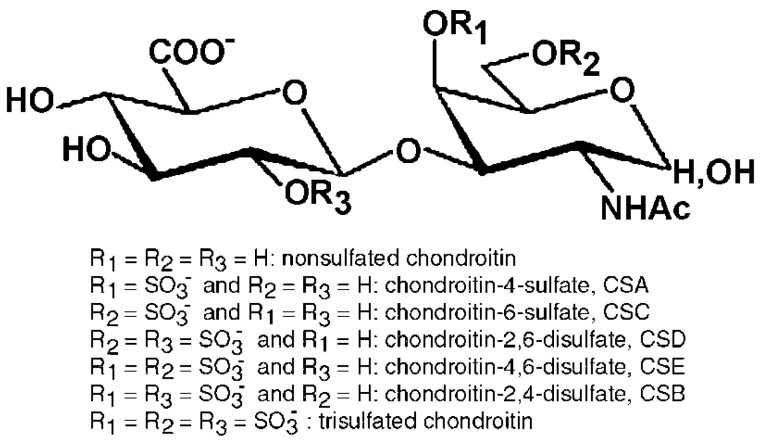
Structure of the disaccharides present in chondroitin sulfate backbone constituted of d-glucuronic acid and *N*-acetyl-d-galactosamine linked by β(1→3) bonds. The various disaccharides are linked each other by β(1→4) linkages. Minor percentages of very rare disaccharides may also have a sulfate group in position C3 of the glucuronic acid. Ac, acetyl group.

**Figure 2 molecules-24-01447-f002:**
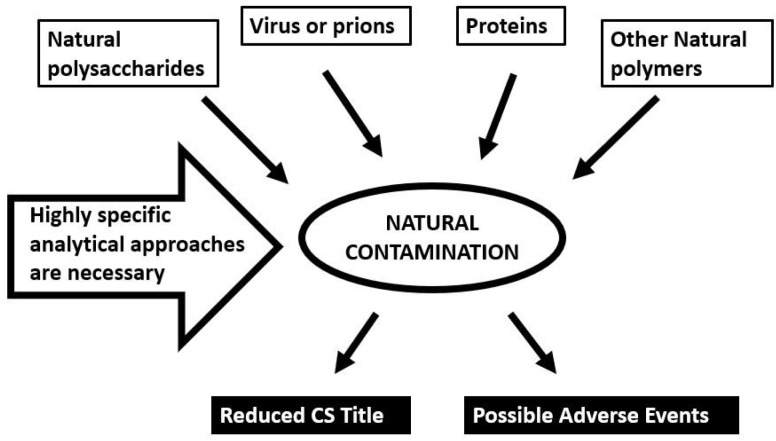
Natural contaminations eventually present in chondroitin sulfate preparations and responsible for reduced title and possible side-effects.

**Figure 3 molecules-24-01447-f003:**
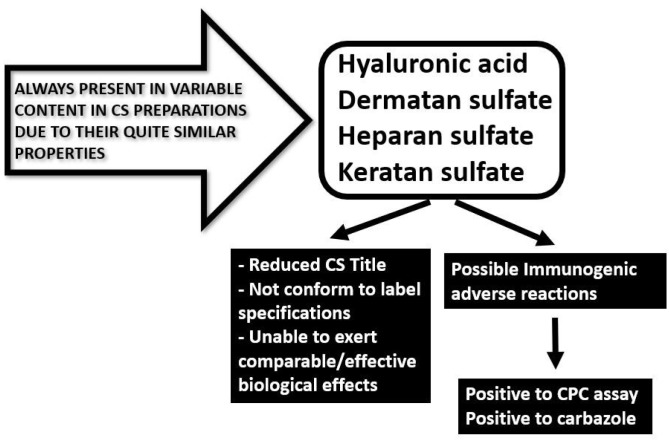
Natural polysaccharides eventually present in chondroitin sulfate preparations and the consequences on the chondroitin quality and title.

**Figure 4 molecules-24-01447-f004:**
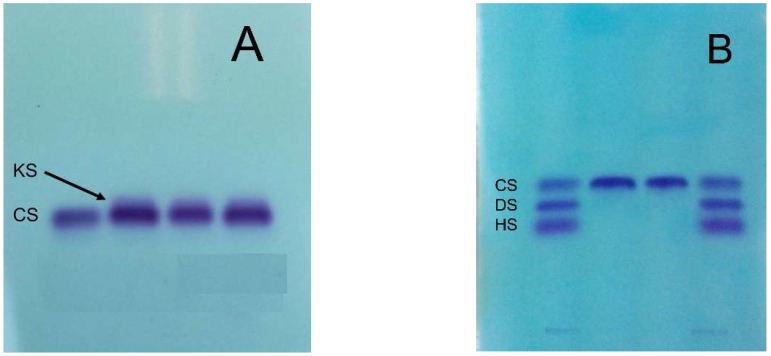
Acetate cellulose electrophoresis able to detect the presence of (**A**) keratan sulfate (KS) or (**B**) dermatan sulfate (DS) and/or heparan sulfate (HS) in chondroitin sulfate (CS) preparations (material owned by the same author).

**Figure 5 molecules-24-01447-f005:**
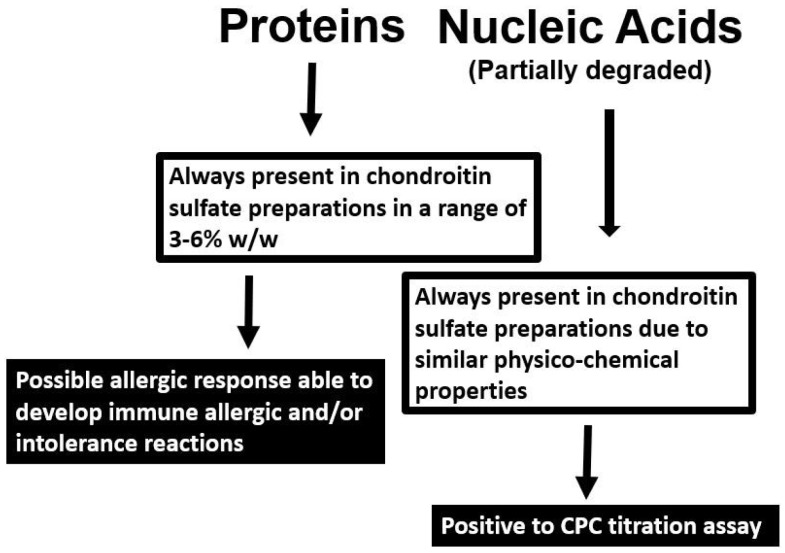
Natural biomolecules possibly present in chondroitin sulfate preparations and their related problems.

**Figure 6 molecules-24-01447-f006:**
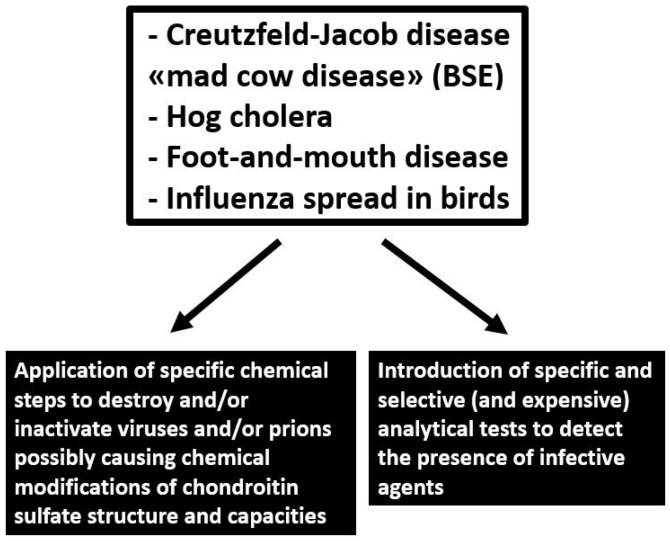
Common possible pathogen contamination and transmissible infective agents in chondroitin sulfate preparations and their effects.

**Figure 7 molecules-24-01447-f007:**
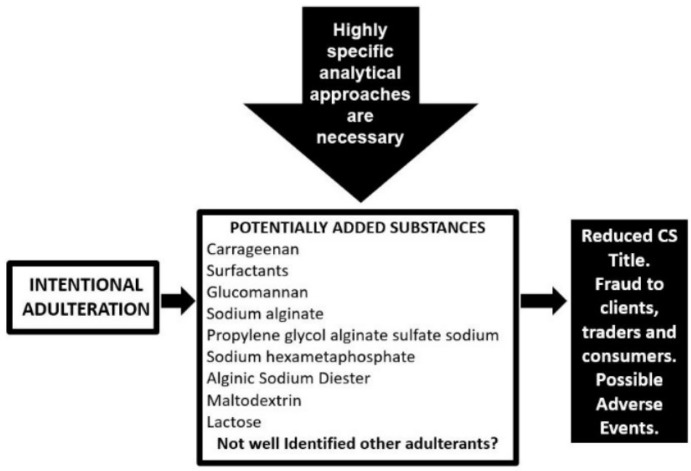
Artificially adulterated chondroitin sulfate by identified substances and their consequences.

**Figure 8 molecules-24-01447-f008:**
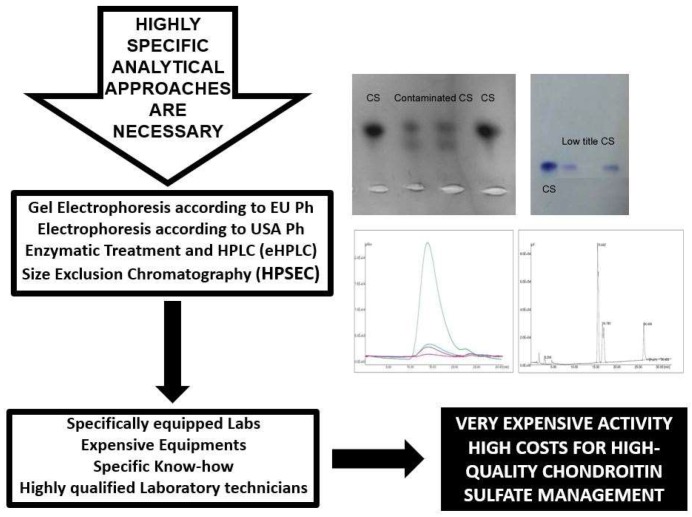
Specific analytical procedures able to distinguish between chondroitin sulfate produced from different origin and to assess CS quality and real quantity.

**Table 1 molecules-24-01447-t001:** Main general characteristics and properties of animal CS for commercial purposes.

Variable and generally not defined and heterogeneous source of extraction
Possible cross-contamination between sources of different origin
Possible presence of bacteria, virus and/or prions
High content of proteins that are not characterized (up to 5–10%). Some of these proteins may have allergenic potential able to develop immune reactions
Variable content of immunogenic keratan sulfate and other natural biopolymers
Variable purity
Heterogeneous structure and physicochemical properties. Variable molecular mass, polydispersity and charge density
Process of extraction generally not controlled causing possible modifications of the CS structure, such as desulfation and/or depolymerization
Possible intentional adulteration by artificial (macro)molecules
Possible batch-to-batch variability
In general, no evaluation of any biological activity

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
