# Peer review of "Chondroitin Sulfate Safety and Quality"

_molecules, 2019, doi:10.3390/molecules24081447_

Round 1
Reviewer 1 Report
Manuscript ID molecules-469600
Title Analytical controls for chondroitin sulfate safety and quality
Comments:
The author reviews possible sources of contamination of chondroitin sulfate, its structural variability due to the diverse raw materials used to obtain CS, and associated production processes to end by outlining analytical methods suitable for quality control. While the topic is relevant, the originality of the work is questionable since similar reviews have been previously published, including some by the author of the present manuscript (see ref. 5, 6 and 26). Furthermore, and despite the title of the work, less than two pages are devoted to the discussion of analytical quality controls of CS. This section should be expanded to provide details about the techniques proposed, their capabilities to identify and quantify each contaminant previously mentioned, as opposed to currently used methods, and establish the origin of the material tested. Also, the manuscript would benefit from further editing, as information is made redundant in some instances and a number sentences sound unnatural and are difficult to understand. The author can find below specific comments:
Lines 29-30: “Apart KS composed of galactose and N-acetyl-D-galactosamine (GalNAc) [3], the backbone of the other GAGs….”.
This sentence is not grammatically correct, it should read “Apart from KS, which is composed of…”
Lines 33-36: “KS, CS/DS and HS/heparin are sulfated macromolecules possessing different degrees of charge density thanks to the presence of sulfate groups in varying amounts and located in different positions. As a consequence, these polysaccharides are very heterogeneous for relative molecular mass, charge density, physico-chemical properties, biological and pharmacological properties”.
While as above stated different proportions of sulfated units influence the molecular weight of CS, chain length is the main factor. The author should rewrite the sentence to clarify this fact. Also, “…for relative molecular mass…” would read better as “…in terms of relative…” Finally, it is unclear why the molecular mass is termed “relative”.
Lines 38-40: The wording of these sentences is confusing and should be rewritten. In its present form it conveys the idea that CS is formed by different disaccharides, hence CS may be biosynthesized based on these differences. Quite the opposite, the CS backbone is sulfated in different positions during biosynthesis leading to a heterogeneous copolymer. Also, besides β(1→3) glycosidic bonds between GlcA and GalNAc (line 38), the author should mention that linkage between GalNAc and GlcA occurs by β(1→4) glycosidic bonds. The same occurs in the caption of Fig. 1. In line 41, a comma should be placed after “respectively” instead of before.
In line 49, “quite always” would be better replaced by “almost always” for instance.
In line 59 the author uses CSB as synonym of DS. The former term was common decades ago, but may lead to confusion as GlcA2S-GalNAc4S is also known as CS-B.
In line 77 “and” should be placed after “organogenesis”. The same in line 79 after “ligaments”.
In line 91 “Countries” should be in lower case.
In line 112 the author states that commercial CS is produced from bony fish and includes a reference to an academic work focused on the characterization of CS from sturgeon. However, to the best of my knowledge, CS is only commercially available from cartilaginous fish. Further references should be included to show its commercial nature.
In line 109 “natural nature” should be rephrased, “fishes” in line 12 is not correct and should be written in singular form, and “possible” in line 112 should be replaced by “possibly”.
Lines 127, 128, 155, 166, 212, and 314 should be revised for other grammatical mistakes.
Sentence in lines 166-168 is confusing and should be rewritten.
The author needs to provide the source of fig. 4
In section 3.1.3., it would be nice if the author could provide references to CS found contaminated with infective agents, as well as how elimination or inactivation of such affects CS structure. In line 241 it is mentioned that prions represent a major concern for pharmaceutical products derived from animals, however in the case of CS this is limited to terrestrial sources. This should be made clear.
Line 268 links CS manufactured inChinato intentional adulteration. Is this fraudulent practice actually related to the manufacturing country or to particular companies? The author should carefully rephrase this sentence, as it could create the impression that Chinese manufacturing is of low quality.
In section 4 the author should relate in detail each contaminant previously outline with each method of analysis. A flowchart with the QC process would greatly help in understanding the necessary steps.
Ref 22., Should be 2011 where it says 1011?
Author Response
Reviewer 1
The author reviews possible sources of contamination of chondroitin sulfate, its structural variability due to the diverse raw materials used to obtain CS, and associated production processes to end by outlining analytical methods suitable for quality control. While the topic is relevant, the originality of the work is questionable since similar reviews have been previously published, including some by the author of the present manuscript (see ref. 5, 6 and 26).
Yes, but this new review add new information, clarify some important points and further focus on this topic that is, also according to the reviewers, very important and still debated. Moreover, reference 26 does not focus attention of CS quality in relation to its purity, possible presence of adulterants and analytical evaluation.
Furthermore, and despite the title of the work, less than two pages are devoted to the discussion of analytical quality controls of CS. This section should be expanded to provide details about the techniques proposed, their capabilities to identify and quantify each contaminant previously mentioned, as opposed to currently used methods, and establish the origin of the material tested.
This paper is a review and not a specific QC procedure or analytical technical paper to apply for the production and analysis of CS. On the other hand, the quoted references and the many scientific papers in literature are available for further analytical details if desired by the readers of Molecules. This review has the aim to illustrate the general feature related to CS safety and quality.
Also, the manuscript would benefit from further editing, as information is made redundant in some instances and a number sentences sound unnatural and are difficult to understand. The author can find below specific comments.
This has been done also thank to the reviewers’ comments.
Lines 29-30: “Apart KS composed of galactose and N-acetyl-D-galactosamine (GalNAc) [3], the backbone of the other GAGs….”. This sentence is not grammatically correct, it should read “Apart from KS, which is composed of…”
Done according to the reviewer’ request.
Lines 33-36: “KS, CS/DS and HS/heparin are sulfated macromolecules possessing different degrees of charge density thanks to the presence of sulfate groups in varying amounts and located in different positions. As a consequence, these polysaccharides are very heterogeneous for relative molecular mass, charge density, physico-chemical properties, biological and pharmacological properties”. While as above stated different proportions of sulfated units influence the molecular weight of CS, chain length is the main factor. The author should rewrite the sentence to clarify this fact. Also, “…for relative molecular mass…” would read better as “…in terms of relative…” Finally, it is unclear why the molecular mass is termed “relative”.
The sentence has been modified according to the reviewer’ request.
Lines 38-40: The wording of these sentences is confusing and should be rewritten. In its present form it conveys the idea that CS is formed by different disaccharides, hence CS may be biosynthesized based on these differences. Quite the opposite, the CS backbone is sulfated in different positions during biosynthesis leading to a heterogeneous copolymer. Also, besides β(1→3) glycosidic bonds between GlcA and GalNAc (line 38), the author should mention that linkage between GalNAc and GlcA occurs by β(1→4) glycosidic bonds.
The sentence has been modified according to the reviewer’ request.
The same occurs in the caption of Fig. 1.
Done.
In line 41, a comma should be placed after “respectively” instead of before.
Done.
In line 49, “quite always” would be better replaced by “almost always” for instance.
Done.
In line 59 the author uses CSB as synonym of DS. The former term was common decades ago, but may lead to confusion as GlcA2S-GalNAc4S is also known as CS-B.
No, indeed. The term CSB or CS-B is referred to DS. GlcA2S-GalNAc4S (and IdoA2S-GalNAc4S) is one of the main disulfated disaccharides forming CS/DS.
In line 77 “and” should be placed after “organogenesis”. The same in line 79 after “ligaments”.
In line 91 “Countries” should be in lower case.
Done according to the reviewer’ request.
In line 112 the author states that commercial CS is produced from bony fish and includes a reference to an academic work focused on the characterization of CS from sturgeon. However, to the best of my knowledge, CS is only commercially available from cartilaginous fish. Further references should be included to show its commercial nature.
No official reference is available as this product is new on the market. “Personal information” has been added to the list of references into the text.
In line 109 “natural nature” should be rephrased, “fishes” in line 12 is not correct and should be written in singular form, and “possible” in line 112 should be replaced by “possibly”.
Done according to the reviewer’ request.
Lines 127, 128, 155, 166, 212, and 314 should be revised for other grammatical mistakes.
Done according to the reviewer’ request.
Sentence in lines 166-168 is confusing and should be rewritten.
Done according to the reviewer’ request.
The author needs to provide the source of fig. 4.
Done according to the reviewer’ request.
In section 3.1.3., it would be nice if the author could provide references to CS found contaminated with infective agents,
a new reference, number 37 has been added.
as well as how elimination or inactivation of such affects CS structure.
This is already reported “possibly introducing chemical modifications in the CS structure such as degradation, desulfation, oxidation of carbohydrate chains, introduction of chemical groups, etc.”
In line 241 it is mentioned that prions represent a major concern for pharmaceutical products derived from animals, however in the case of CS this is limited to terrestrial sources. This should be made clear.
Done according to the reviewer’ request.
Line 268 links CS manufactured in China to intentional adulteration. Is this fraudulent practice actually related to the manufacturing country or to particular companies? The author should carefully rephrase this sentence, as it could create the impression that Chinese manufacturing is of low quality.
I disagree with the interpretation of the referee as the sentence just reports that 80-90% of CS is produced in China (and this is true) and not that the 80-90% of CS produced in China is intentionally adultered. Anyway, this sentence has been changed in “…and intentional adulteration is possibly practiced by the nutraceutical manufacturers of all over the word with the aim…”.
In section 4 the author should relate in detail each contaminant previously outline with each method of analysis.
I disagree with this request as this paper is a review and not a specific technical note or procedure describing the method of analysis related to each known (and unknown) CS contaminant.
A flowchart with the QC process would greatly help in understanding the necessary steps.
I disagree with this request as this paper is a review and not a specific QC procedure to apply for the production and analysis of CS. Moreover, as largely illustrated in the review, each Company uses different procedures and analytical approaches to assess the quality of its product and this is object of discussion.
Ref 22., Should be 2011 where it says 1011?
Yes, corrected.
Reviewer 2 Report
The manuscript entitled « Analytical controls for chondroitin sulfate safety and quality » aims at presenting analytics such as electrophoresis, eHLPC and HPSec to control the quality of pharmaceutical chondroitin sulfate. The number of cited methods seems limited finally; which ones are implemented in the industry? The manuscript is sometimes confusing and not precise enough and some examples and explanations are lacking. More details are given below and can eventually help in improving the manuscript.
At the beginning the structural diversity of CS is reviewed followed by the biological activities of CS, especially in articular and bone metabolism.
Q1 : Since chemical structure of CS is very diverse, is there any data on structure-function relationship ?. In other terms do structural variants have different biological activities ? Are there some CS forms less active or even toxic? It is a major issue in the production of drugs from natural products and for the control of the drug quality.
The core of the manuscript (Safety and quality concerns) begins with sources from which CS are extracted. The diversity of sources for extraction results in structural diversity, and in the risk to co-extract pathogens or other bioactive molecules considered as natural contaminants. Some other compounds may also be added upon drug manufacturing.
Q2 : I am not sure Table 1 is necessary since it gives nothing new : the text has already presented the ideas. Furthermore, this table is only a list of issues : they are not well organized (grouped by idea for example…) and there is only one column, not enough to be a table.
The following paragraphs deal with charge and molecular weight importance in the biological activity and pharmacokinetics, with structural modifications due to extraction and purification process. More details are given on natural contaminants in 3.1. Polysaccharidic contaminants are detected by ELISA and electrophoresis is presented in the case of HA, DS and KS.
Q3 : Some information are lacking in Figure 4 such as the differences in the preparations, the staining, the difference between A and B (are they the same preparations ?). Are these electrophoreses extracted from the cited littérature ?
Other contaminating biomolecules are reviewed : nucleic acids and proteins.
Q4 : References are not enough informative because only one is proposed and the protocols of purification are not explained. What could be the improvement to get rid of these contaminants ? Ref 31 deals with capillary electrophoresis which could be an analytic method useful for the quality control. It is not cited or discussed in the manuscript.
Q5 : What is CPC assay ? I don’t think it has been explained before. (line 224)
Q6 : Concerning infective agents, could you please provide any references (lines 222-235) ? the consequences on extraction processes and the risks related to these agents are presented. Detection is also discussed but not precise enough : line 252, what is this diagnostic test ?
Q7 : In the paragraph on intentional adulterations, in Line 259, what are these analytical controls used in the nutraceutical industry ? Figure 7 : These substances are not « artificial », please replace by « potentially added ».
Q8 : The paragraph between lines 277 to 290 should be in the analytic control one (number 4).
Q9 : Line 298 : why « ? »
Q10 : Old analytical quality control methods are presented together with newer ones. But more explanations should be necessary : what is CPC titration, eHPLC (which enzymes, detailed presentation should be given at line 316 and not at line 334). Then more specific techniques are necessary to identify the origin : but again they are HPSec and eHPLC…
Q11 : Line 345 : you write analytical techniques are illustrated but I think precise illustration, explanation of analytics are lacking (principle, applications) and a specific paragraph should better be proposed on technique protocols.
Q12 : Lines 354-358 : what are the chemical structures of these standards ?
Q13 : Lines 146, 162-163, 169, : the author writes about the extraction processes : their varieties, their time length and complexity, additional purification sometimes necessary, high costs… But none of these processes have been detailed. Some examples should be added.
English language to be checked :
Line 15 : may lead (instead of may led)
Line 16 : surely instead of sure
Line 18-19 : the sentence is not clear
Line 35 : « heterogeneous for » does not sound correct
Line 65-67 : please check the sentence
Line 74 : consequently
Line 130 : « heterogeneous for » does not sound correct
Table 1 : desulfation (not desulfatation)
Line 135 : repetition of have-having.
Line 148 : undesired instead of undesidered
Lines 156-159 : the sentence is complex and not clear enough
Line 272 : I don’t understand « minimize » in this context
Lines 301 : no more instead of not more
Line 302 : non specificity, unspecificity ?
Line 307 : please add « measurement » after « intrinsic viscosity »
Line 314 : Replace « related to » by « encounetered in »
Line 338 : I don’t understand « is used to its production or cross-contamination is possible »
Line 361-364 : please check the sentence : complex, what is iv ?
Author Response
Reviewer 2
The manuscript entitled « Analytical controls for chondroitin sulfate safety and quality » aims at presenting analytics such as electrophoresis, eHLPC and HPSec to control the quality of pharmaceutical chondroitin sulfate. The number of cited methods seems limited finally; which ones are implemented in the industry? The manuscript is sometimes confusing and not precise enough and some examples and explanations are lacking. More details are given below and can eventually help in improving the manuscript.
At the beginning the structural diversity of CS is reviewed followed by the biological activities of CS, especially in articular and bone metabolism.
Q1 : Since chemical structure of CS is very diverse, is there any data on structure-function relationship ?. In other terms do structural variants have different biological activities ? Are there some CS forms less active or even toxic? It is a major issue in the production of drugs from natural products and for the control of the drug quality.
This aspect, the activity in relation with CS structure and quality is largely mentioned through the entire paper (see in particular paragraph 3). On the other hand, this not a specific aim of this review.
The core of the manuscript (Safety and quality concerns) begins with sources from which CS are extracted. The diversity of sources for extraction results in structural diversity, and in the risk to co-extract pathogens or other bioactive molecules considered as natural contaminants. Some other compounds may also be added upon drug manufacturing.
Q2 : I am not sure Table 1 is necessary since it gives nothing new : the text has already presented the ideas. Furthermore, this table is only a list of issues : they are not well organized (grouped by idea for example…) and there is only one column, not enough to be a table.
Even if some of the aspects reported in the Table are in the text, I believe that Table 1 is important as immediately gives to the readers of the Journal the real situation of CS properties as it groupes all the main charactristics of commercially available CS.
The following paragraphs deal with charge and molecular weight importance in the biological activity and pharmacokinetics, with structural modifications due to extraction and purification process. More details are given on natural contaminants in 3.1. Polysaccharidic contaminants are detected by ELISA and electrophoresis is presented in the case of HA, DS and KS.
Q3 : Some information are lacking in Figure 4 such as the differences in the preparations, the staining, the difference between A and B (are they the same preparations ?). Are these electrophoreses extracted from the cited littérature?
Some more details are reported in the Figure legend. On the other hand, this paper is a review and not a specific technical note or procedure describing the method of analysis related to electrophoresis.
Other contaminating biomolecules are reviewed : nucleic acids and proteins.
Q4 : References are not enough informative because only one is proposed and the protocols of purification are not explained. What could be the improvement to get rid of these contaminants?
This paper is a review and not a specific QC procedure to apply for the production and analysis of CS. Moreover, as largely illustrated in the review, each Company uses different procedures and analytical approaches to assess the quality of its product and this is object of discussion.
Ref 31 deals with capillary electrophoresis which could be an analytic method useful for the quality control. It is not cited or discussed in the manuscript.
Yes, because capillary electrophoresis is generally not used for quality control contrary to HPLC as well as very few of this technique is in the various Pharmacopeias. On the other hand, there are many other analytical methods useful to the valutation of CS structure and properties but this paper is a review on CS quality and not on the analytical approaches for CS.
Q5 : What is CPC assay ? I don’t think it has been explained before. (line 224)
CPC is cetylpyridinium chloride as reported before on line 211.
Q6 : Concerning infective agents, could you please provide any references (lines 222-235) ? the consequences on extraction processes and the risks related to these agents are presented. Detection is also discussed but not precise enough : line 252, what is this diagnostic test ?
There are several diagnostic tests but, as above, this paper is a review and not a specific QC procedure to apply for the production and analysis of CS.
Q7 : In the paragraph on intentional adulterations, in Line 259, what are these analytical controls used in the nutraceutical industry ?
This is largely reported and discussed in this revies, in particulat the non-specific analytical methods such as CPC and carbazole assay.
Figure 7 : These substances are not « artificial », please replace by « potentially added ».
Modified according to the reviewer’ request.
Q8 : The paragraph between lines 277 to 290 should be in the analytic control one (number 4).
I disagree with the reviewer as lines 277-290 are referred to the argument of this paragraph, number 3.2, related to intentional adulteratiions and related substances.
Q9 : Line 298 : why « ? »
The sentence and the question have been clarified.
Q10 : Old analytical quality control methods are presented together with newer ones. But more explanations should be necessary : what is CPC titration, eHPLC (which enzymes, detailed presentation should be given at line 316 and not at line 334).
This review reports “On the contrary, agarose-gel electrophoresis according to European pharmacopoeia, electrophoresis on acetate of cellulose (ACE) according to US pharmacopoeia, HPSEC and eHPLC according to the AOAC (Association of Official Analytical Chemists) [49] are highly specific analytical approaches for the determination of CS quality, quantity, chemical properties and structure.” discussing the differences between the classic aspecific CPC analysis used in Companies compared to the more modern and specific analytical approaches. On the other hand, this paper is a review and not a specific QC procedure or analytical technical paper to apply for the production and analysis of CS. Finally, the quoted references and the many scientific papers in literature are available for further analytical details if desired by the readers of the Molecules. This review has the aim to illustrate the general feature related to CS safety and quality.
Then more specific techniques are necessary to identify the origin : but again they are HPSec and eHPLC…
As above, this paper is a review and not a specific QC procedure or analytical technical paper to apply for the production and analysis of CS. On the other hand, the quoted references and the many scientific papers in literature are available for further analytical details if desired by the readers of the Molecules. This review has the aim to illustrate the general feature related to CS safety and quality.
Q11 : Line 345 : you write analytical techniques are illustrated but I think precise illustration, explanation of analytics are lacking (principle, applications) and a specific paragraph should better be proposed on technique protocols.
I disagree with this request as this paper is a review and not a specific QC procedure or analytical technical paper to apply for the production and analysis of CS. On the other hand, the quoted references and the many scientific papers in literature are available for further analytical details if desired by the readers of the Molecules. This review has the aim to illustrate the general feature related to CS safety and quality.
Q12 : Lines 354-358 : what are the chemical structures of these standards ?
If necessary, the readers of Molecules can go to the illustrated links and quoted references for more details related to available CS standards.
Q13 : Lines 146, 162-163, 169, : the author writes about the extraction processes : their varieties, their time length and complexity, additional purification sometimes necessary, high costs… But none of these processes have been detailed. Some examples should be added.
This paper is a review and not a specific QC procedure to apply for the production and analysis of CS. Moreover, as largely illustrated in the review, each Company uses different procedures and analytical approaches to assess the quality of its product and this is object of discussion.
English language to be checked :
Line 15 : may lead (instead of may led)
OK
Line 16 : surely instead of sure
OK..
Line 18-19 : the sentence is not clear
This sentence has been modified.
Line 35 : « heterogeneous for » does not sound correct
Modified.
Line 65-67 : please check the sentence
This sentence has been modified.
Line 74 : consequently
OK.
Line 130 : « heterogeneous for » does not sound correct
OK.
Table 1 : desulfation (not desulfatation)
OK
Line 135 : repetition of have-having.
OK..
Line 148 : undesired instead of undesidered
OK.
Lines 156-159 : the sentence is complex and not clear enough
This sentence has been modified.
Line 272 : I don’t understand « minimize » in this context
Changed with reduce.
Lines 301 : no more instead of not more
OK.
Line 302 : non specificity, unspecificity ?
OK.
Line 307 : please add « measurement » after « intrinsic viscosity »
OK.
Line 314 : Replace « related to » by « encounetered in »
Modified.
Line 338 : I don’t understand « is used to its production or cross-contamination is possible »
This sentence has been modified.
Line 361-364 : please check the sentence : complex, what is iv ?
This sentence has been modified.
Reviewer 3 Report
Author reports an analytical investigation of chondroitin sulfate (CS) in animal tissue materials derived from different terrestrial or marine species. The work is well done, however there are some important references published recently about chemical and spectroscopic properties of diferent chondroitin sulate compounds authors has not cited. For example, the work of Cunha et al. (Carbohydrate Polymers, 134, 2015, 300-308, http://dx.doi.org/10.1016/j.carbpol.2015.08.006) is not cited, and in that investigation authors have described the analysis of different structures for CS involving 4-sulfated:6-sulfated disaccharides, which varies from 0.9 to 1.7. Is that important or not in the description of the analytical approach? As author cites, there are several problems with frauds involving CS; is there any suggestion of procedures to be done to avoid that? In the paper cited above, authors have provided an extensive investigation, showing that from 16 commercial samples around the world, only 5 presented more than 90% of the label description of CS. Is there anything to be done on this? Figure 8 shows an overview about the techniques which can be used for analytical detection of CS; once again, the use of Raman together the use of agarose gel electrophoresis, as well as FACE to determine both the CS contents and the eventual presence/identification of contaminants, is recommended, according Cunha and co-workers. This is not exactly what author cites in his conclusions. Please present a discussion. Table 1 is unuseful, and should be written in the text. References should be revised, mainly the ones containing the chemical analyses of CS. As an overview, the manuscript is a nice contribution, since the points are properly addressed.Author Response
Reviewer 3
Author reports an analytical investigation of chondroitin sulfate (CS) in animal tissue materials derived from different terrestrial or marine species. The work is well done, however there are some important references published recently about chemical and spectroscopic properties of diferent chondroitin sulate compounds authors has not cited.
For example, the work of Cunha et al. (Carbohydrate Polymers, 134, 2015, 300-308, http://dx.doi.org/10.1016/j.carbpol.2015.08.006) is not cited, and in that investigation authors have described the analysis of different structures for CS involving 4-sulfated:6-sulfated disaccharides, which varies from 0.9 to 1.7. Is that important or not in the description of the analytical approach?
This paper has been quoted in reference list and discussed in paragraph 3 and
As author cites, there are several problems with frauds involving CS; is there any suggestion of procedures to be done to avoid that?
Yes, and this is largely reported into the paper, the use of specific analytical approaches such as those illustrated in paragraph 4.
In the paper cited above, authors have provided an extensive investigation, showing that from 16 commercial samples around the world, only 5 presented more than 90% of the label description of CS. Is there anything to be done on this?
Yes, and this is largely reported into the paper, the use of specific analytical approaches such as those illustrated in paragraph 4.
Figure 8 shows an overview about the techniques which can be used for analytical detection of CS; once again, the use of Raman together the use of agarose gel electrophoresis, as well as FACE to determine both the CS contents and the eventual presence/identification of contaminants, is recommended, according Cunha and co-workers. This is not exactly what author cites in his conclusions. Please present a discussion.
These aspects have been introduced and shortly discussed also considering that this is a review having the aim to give a general overview and reporting specific scientific papers in reference list useful for further possible information and analytical details.
Table 1 is unuseful, and should be written in the text.
Even if some of the aspects reported in the Table are in the text, I believe that Table 1 is important as immediately gives to the readers of the Journal the real situation of CS propeties as it groupes all the main charactristics of commercially available CS.
References should be revised, mainly the ones containing the chemical analyses of CS.
Reference list has been modified by adding more references.
As an overview, the manuscript is a nice contribution, since the points are properly addressed.
OK.
Reviewer 4 Report
The present "Review" might become very interesting and useful if the author included some data published by other authors, besides his own data (out of the 50 citations here included, 11 are auto-citations, corresponding to 22%).
This reviewer is aware that the author is an expert, an authority, in the area, but there are also other groups giving important contributions to the field.
Just to cite a few examples of representative papers recently published, which were neglected:
1. A study on the structure of different chondroitin sulfate commercial preparations, as well as identification of the contaminants in these preparations has appeared in 2015 (Cunha et al., Carbohydrate Polymers 134 (2015) 300-308). This study has shown that, out of 16 chondroitin sulfate samples prepared in different countries all over the world, only 5 really contained chondroitin sulfate and were in conformity to their labels. The remaining 11 samples contained maltodextrin or lactose as the main components (adulterations?).
2. Stabler et al. have also shown that chondroitin sulfate inhibits NF-kB activity (Osteoarthr. Cartilage 25 (2017) 166–174).
3. Other paper, published in 2017, reported that chondroitin sulfates with different structures have anti-inflammatory activities on macrophages and chondrocytes (Cunha et al., Int. J. Biol. Macromol. 103 (2017) 1019-1031).
This reviewer suggests the inclusion of papers published by other groups to further improve the quality of the present text.
Author Response
Reviewer 4
The present "Review" might become very interesting and useful if the author included some data published by other authors, besides his own data (out of the 50 citations here included, 11 are auto-citations, corresponding to 22%). This reviewer is aware that the author is an expert, an authority, in the area, but there are also other groups giving important contributions to the field.
Done according to the reviewer’ request.
Just to cite a few examples of representative papers recently published, which were neglected:
1. A study on the structure of different chondroitin sulfate commercial preparations, as well as identification of the contaminants in these preparations has appeared in 2015 (Cunha et al., Carbohydrate Polymers 134 (2015) 300-308). This study has shown that, out of 16 chondroitin sulfate samples prepared in different countries all over the world, only 5 really contained chondroitin sulfate and were in conformity to their labels. The remaining 11 samples contained maltodextrin or lactose as the main components (adulterations?).
This paper has been quoted and reported in the paper.
2. Stabler et al. have also shown that chondroitin sulfate inhibits NF-kB activity (Osteoarthr. Cartilage 25 (2017) 166–174).
This paper is related to a specific biological activity of CS and it is out of the aim of this review. Moreover, a paragraph on the CS activity, par. 2, is already present in the review showing its main properties.
3. Other paper, published in 2017, reported that chondroitin sulfates with different structures have anti-inflammatory activities on macrophages and chondrocytes (Cunha et al., Int. J. Biol. Macromol. 103 (2017) 1019-1031).
Also this paper is related to a specific biological activity of CS and it is out of the aim of this review. Moreover, a paragraph on the CS activity, par. 2, is already present in the review showing its main properties.
This reviewer suggests the inclusion of papers published by other groups to further improve the quality of the present text.
This has been done where possible.
Round 2
Reviewer 1 Report
While minor changes have been made, the major point has not been addressed.
Author Response
Yes, however, thank to the extensive modifications made, the paper has been deeply revised and strongly improved.
Reviewer 2 Report
As written in the responses, the author does not wish to provide details on quality control procedures. I can understand this point of view ; in that case I find that "Analytical controls" is not appropriate in the title and in the abstract (line 20). "Chondroitin sulfate safety and quality" might be better.
There are some little misspellings:
line 133 "possibly" instead of "possiby"
line 200 "more steps" instead of "more step"
line 407: "mass spectrometry" instead of "mass spectroscopy"
Figure 5: "allergic" instead of allergenic
Author Response
As written in the responses, the author does not wish to provide details on quality control procedures. I can understand this point of view ; in that case I find that "Analytical controls" is not appropriate in the title and in the abstract (line 20). "Chondroitin sulfate safety and quality" might be better.
Thank to the reviewer for his/her comprehension. I agree to change the title in "Chondroitin sulfate safety and quality".
There are some little misspellings:
line 133 "possibly" instead of "possiby"
Done.
line 200 "more steps" instead of "more step"
Done.
line 407: "mass spectrometry" instead of "mass spectroscopy"
Done.
Figure 5: "allergic" instead of allergenic
Done.